# A Comprehensive Overview of Globally Approved JAK Inhibitors

**DOI:** 10.3390/pharmaceutics14051001

**Published:** 2022-05-06

**Authors:** Ahmed M. Shawky, Faisal A. Almalki, Ashraf N. Abdalla, Ahmed H. Abdelazeem, Ahmed M. Gouda

**Affiliations:** 1Science and Technology Unit (STU), Umm Al-Qura University, Makkah 21955, Saudi Arabia; amesmail@uqu.edu.sa; 2Department of Pharmaceutical Chemistry, Faculty of Pharmacy, Umm Al-Qura University, Makkah 21955, Saudi Arabia; 3Department of Pharmacology and Toxicology, Faculty of Pharmacy, Umm Al-Qura University, Makkah 21955, Saudi Arabia; anabdrabo@uqu.edu.sa; 4Department of Pharmacology and Toxicology, Medicinal and Aromatic Plants Research Institute, National Center for Research, Khartoum 2404, Sudan; 5Department of Medicinal Chemistry, Faculty of Pharmacy, Beni-Suef University, Beni-Suef 62514, Egypt; ahmed.haasn77@yahoo.com; 6Department of Pharmacy, College of Pharmacy, Riyadh Elm University, Riyadh 11681, Saudi Arabia

**Keywords:** JAK, synthesis, kinase inhibitory activity, pharmacological uses, binding mode/interactions

## Abstract

Janus kinase (JAK) is a family of cytoplasmic non-receptor tyrosine kinases that includes four members, namely JAK1, JAK2, JAK3, and TYK2. The JAKs transduce cytokine signaling through the JAK-STAT pathway, which regulates the transcription of several genes involved in inflammatory, immune, and cancer conditions. Targeting the JAK family kinases with small-molecule inhibitors has proved to be effective in the treatment of different types of diseases. In the current review, eleven of the JAK inhibitors that received approval for clinical use have been discussed. These drugs are abrocitinib, baricitinib, delgocitinib, fedratinib, filgotinib, oclacitinib, pacritinib, peficitinib, ruxolitinib, tofacitinib, and upadacitinib. The aim of the current review was to provide an integrated overview of the chemical and pharmacological data of the globally approved JAK inhibitors. The synthetic routes of the eleven drugs were described. In addition, their inhibitory activities against different kinases and their pharmacological uses have also been explained. Moreover, their crystal structures with different kinases were summarized, with a primary focus on their binding modes and interactions. The proposed metabolic pathways and metabolites of these drugs were also illustrated. To sum up, the data in the current review could help in the design of new JAK inhibitors with potential therapeutic benefits in inflammatory and autoimmune diseases.

## 1. Introduction 

Janus kinases (JAKs) are intracellular, non-receptor tyrosine kinases [1]. The JAK family consists of four members, including JAK1, JAK2, JAK3, and TY2K. Since the first discovery of JAKs by Wilks thirty years ago [2], great efforts have been made to understand their structure and functions. The four JAKs play an essential role in the transduction of the cytokine-mediated signals, which takes place through the JAK-signal transducers and activators of the transcription (STAT) pathway [3]. Four members of the JAK family have emerged as potential drug targets in different types of diseases [4].

In the current review, we aimed to provide an integrated overview of the chemical and pharmacological data of the globally approved JAK inhibitors that were approved for the treatment of inflammatory, autoimmune, and myeloproliferative diseases. Highlighting the differences in chemical structure, binding interactions, kinase inhibitory activities, pharmacological activities, and metabolic pathways of these drugs could help in the design of new, more potent, and safer JAK inhibitors. 

### 1.1. Structure of JAKs

The structure of JAKs (Figure 1) consists of 7 domains, JH1-JH7 [5]. The four JAKs have the same domains with an overall similarity of 48% [6]. The first domain is JH1, which exists at the C-terminal. This domain is also called the kinase domain because it is responsible for the enzymatic activity of the kinase. The second domain, JH2, is the pseudo-kinase domain, which lacks the tyrosine kinase activity [7]. However, JH2 plays an important role in the regulation of kinase activity [8]. JAKs also include two domains, JH3-JH4, which share homology with the Src-homology-2 (SH2) domain, while the fourth region of JAKs is the FERM domain, which exists at the N-terminal and plays a role in the binding of JAKs with cytokine receptors [9].

### 1.2. JAK-STAT Pathway

Several ligands such as cytokines and growth factors have been reported to activate the JAK-STAT pathway [10]. Following the activation of JAKs, phosphorylation and dimerization of STATs take place (Figure 2). The phosphorylated STATs enter the nucleus, where they initiate a suitable transcriptional response in the genes that regulate immunity, inflammation, and hematopoiesis [10,11,12].

### 1.3. Therapeutic Potential of JAK Inhibition

The JAK-STAT pathway is responsible for the transduction of cytokines and growth factor signals that play a crucial role in inflammation and autoimmune diseases [13]. Among these kinases, JAK1 seems to play a crucial role in pruritic dermatitis [14], allergic rhinitis [15], asthma [16], and inflammatory bowel disease [17]. Targeting JAK1 with small-molecule inhibitors has proved efficacy in the treatment of these diseases [16,18,19]. Moreover, several small molecules with JAK1 and JAK2 inhibitory activity have also provided therapeutic benefits in the treatment of rheumatoid arthritis, psoriasis, and pruritis [20]. In addition, several JAK3 selective inhibitors have been evaluated for their efficacy in the treatment of rheumatoid arthritis [20]. In addition, JAK1/TYK2 dual inhibitors have also been developed with potential therapeutic options in inflammatory diseases [21]. Furthermore, TYK2 selective inhibitors may also be useful in the treatment of autoimmune diseases [20,22].

On the other hand, excessive activation of JAKs has also been reported in different types of cancer [23]. The JAK/STAT3 pathway plays an important role in the proliferation and angiogenesis of solid tumors [23]. In 2005, the discovery of JAK2 mutation (JAK2V617F) in myeloproliferative neoplasms has attracted much attention [24,25]. This discovery led to a better understanding of these diseases. In addition, JAK2V617F has emerged as a potential therapeutic target for myeloproliferative neoplasms [26,27]. In addition, the JAK1/JAK2 inhibitor, ruxolitinib, has also been approved for the treatment of myelofibrosis and polycythemia vera [28,29]. Moreover, the dual inhibition of JAK2 and FLT3 could also provide therapeutic option in the treatment of acute myelogenous leukemia and myeloproliferative neoplasms [30,31,32].

### 1.4. Classification of JAK Inhibitors

JAK inhibitors can be divided into two generations [33]. The first-generation includes small molecules such as baricitinib and tofacitinib, which act as non-selective inhibitors of JAKs. On the other hand, second-generation drugs such as filgotinib and upadacitinib have selective inhibitory activity against JAKs [33]. This difference in the selectivity of the two generations is associated with some differences in their safety and efficacy.

One the other hand, JAK inhibitors may also be classified based on their binding mode and the type of interactions with the amino acids in JAKs into reversible (competitive) and irreversible (covalent) inhibitors (Figure 3). 

#### 1.4.1. Reversible JAK Inhibitors

Competitive JAK inhibitors form reversible (non-covalent) binding interactions with the amino acids in the four JAKs. The binding interactions formed by this type of JAK inhibitors include hydrogen bonds and hydrophobic interactions. The class of reversible JAK inhibitors can also be classified into two sub-classes. 

##### ATP-Competitive Inhibitors

The mechanism of action of these inhibitors depends on their competition with ATP for the catalytic ATP-binding site in JAKs [27,34]. These inhibitors may also be classified based on the conformation of the kinase domain to which they bind:

Type I JAK Inhibitors

These inhibitors bind to the ATP-binding site of the JAKs under the active conformation of the kinase domain [27,34]. This includes clinically approved drugs such as filgotinib, which acts and is classified as a selective JAK1, while fedratinib exhibits the selective inhibition of JAK2 [20,35]. On the other hand, tofacitinib and peficitinib act by blocking multiple JAKs [20,36]. The ability of type I JAK inhibitors to bind to multiple kinases and act as non-selective inhibitors could be due to the highly conserved structure of the ATP-binding site in the four JAKs [18].

Type II JAK Inhibitors

Type II JAK inhibitors also bind to the ATP-binding site of the kinase domain in the inactive conformation of JAKs [27,34]. NVP-BBT594 and NVP-CHZ868 (Figure 4) are representative examples of type II inhibitor, which target JAK2 [37,38].

Allosteric JAK Inhibitors

The allosteric JAK inhibitors (Figure 5) include small molecule inhibitors that bind to a site other than the ATP-binding site in JAKs [27,34]. Among these inhibitors, deucravacitinib (BMS-986165) act as a selective allosteric inhibitor of TYK2 [39]. In addition, LS104, and ON044580 are examples of JAK2 allosteric inhibitors [34,40,41,42].

#### 1.4.2. Irreversible JAK3 Inhibitors

This class of irreversible JAK inhibitors (Figure 6) that target JAK3 was also reported [43,44,45]. The mechanism of action of these inhibitors depends on the covalent interaction with the unique Cys909 residue in JAK3 [43]. The chemical structure of these inhibitors has a covalent-bond forming group such as acrylamide and α-cyanoacrylamide, which can bind covalently with Cys909 residue (Figure 6).

Compound **1** was reported among a series of cyanamide-based JAK3 covalent inhibitors [43]. Moreover, compounds **2** (IC_50_ = 0.003 μM) and **3** (IC_50_ = 154 pM) were also reported with potent and selective inhibitory activity against JAK3 [44,45]. 

Ritlecitinib (Figure 6) is an irreversible inhibitor of JAK3, which is currently under evaluation in clinical trials in humans [43,46]. The inhibitory activity of ritlecitinib was mediated by a covalent interaction with Cys909 residue in JAK3 [43]. The binding mode and interaction of ritlecitinib in JAK3 (pdb: 5TOZ) are visualized in Figure 7. The crystal structure was downloaded from the protein data bank (https://www.rcsb.org/, accessed on 11 November, 2021). In this work, the crystal structures of JAK kinases were visualized using the Discovery Studio Visualizer [47]. The crystals were prepared by removing water molecules following the previous reports [48,49]. Moreover, 2/3D binding modes of the JAK inhibitors into the co-crystallized kinases were visualized according to the previous report [50].

### 1.5. Mutation of JAKs

Besides the role of wild-type of JAKs in autoimmune and inflammatory diseases, JAK mutants also play crucial roles in myeloproliferative, lymphomas, and leukemias diseases [6]. JAK1 mutation has been found in acute lymphoblastic leukemia [51]. In addition, JAK2 mutation (JAK2V617F) has been associated with the incidence of myeloproliferative neoplasms [24,52]. The occurrence of JAK2 mutation was reported in nearly all patients with polycythaemia vera (PV) [53].

Moreover, the expression of the JAK3 mutant was associated with the induction of leukemia in mice [54]. JAK3 mutation was also reported in small percent of patient with T-cell acute lymphoblastic leukemia. It was also associated with the occurrence of severe combined immunodeficiency [55].

On the other hand, TYK2 mutations were also associated with immunodeficiency with T-cell lymphopenia [56]. It was also associated with increased susceptibility to bacterial and/or viral infections. The deficiency of TYK2 was associated with recurrent respiratory infection [57].

### 1.6. Crystal Structure of JAKs

The advances in X-ray crystallography have provided a lot of data about the structure and function of different members of the JAK family. This data has helped in the design of selective JAK inhibitors [58]. Many of the crystal structures of different JAKs are available from the protein data bank (https://www.rcsb.org/). Visualization of the binding modes and interactions of the different small molecules inhibitors into the four JAKs was done using the Discovery Studio Visualizer [47].

The binding mode and interactions of KEV, a pyrazolopyridine inhibitor with JAK1 kinase (pdb: 6N7A) [19], were visualized in Figure 8. KEV displays hydrogen bond interactions with Glu957 and Leu959 in the hinge region of JAK1.

In addition, many small molecules of divers chemical nature were reported as co-crystallized ligands with JAK2 [59,60]. Among these ligands, IZA is an isoquinoline derivative with potent and pan-JAK inhibitory activity. The binding interactions of IZA show two conventional hydrogen bonds with Leu932 and Glu930 in the hinge region in JAK2 (Figure 8).

The crystal structure of JAK3 (pdb: 3PJC) in complex with PJC, a pyrazolopyridine inhibitor [61] was visualized in Figure 9. PJC shows hydrogen bond interactions with Leu905 and Glu903. On the other hand, the crystal structure of TYK2 kinase (pdb: 6VNX) in complex with R4V, a pyrimidine inhibitor [62] was also visualized in Figure 9. R4V shows one conventional hydrogen bond with Asp1041.

### 1.7. JAK Inhibitors Approved for Clinical Use

In 1996, Meydan et al. reported AG-490 (Figure 10), a JAK2 inhibitor with antileukemic activity [63]. Following this discovery, great efforts have been made in the last two decades to develop potent and selective JAK inhibitors.

In 2011, the FDA-approved ruxolitinib (Figure 10) became the first JAK inhibitor [64]. One year later, tofacitinib was also approved for the treatment of rheumatoid arthritis [64]. In 2017, baricitinib was approved by the FDA for rheumatoid arthritis [65]. However, in 2019, three JAK inhibitors were approved for clinical use [66,67,68]. These drugs include fedratinib and upadacitinib, which were approved by the FDA, whereas peficitinib had already been approved in Japan for rheumatoid arthritis. In 2020, delgocitinib and filgotinib were also approved in Japan for the treatment of atopic dermatitis and rheumatoid arthritis, respectively [69,70].

#### 1.7.1. Abrocitinib

##### Approval History

Abrocitinib (Figure 11) is a JAK1 kinase inhibitor [71,72]. It was approved by the FDA in January 2020 for the treatment of refractory, moderate-to-severe atopic dermatitis [72].

##### Synthesis

Vazquez et al. [59] have reported the synthesis of abrocitinib from cyclobutyl carbamate **4**, Figure 1. The ketone group in **4** was condensed with methylamine and reduced with sodium borohydride to give **5**. The coupling of **6** with the *cis*-isomer of **5** afforded **7**, which underwent the acid-catalyzed hydrolysis of the carbamate group to give **8**. The base-catalyzed sulfonylation of **8** with propane-1-sulfonyl chloride afforded **9**, which underwent deprotection to remove the tosyl moiety and give abrocitinib.

Connor et al. [71] have also developed a commercial route for the synthesis of abrocitinib. The synthesis takes place through a nitrene-type rearrangement, Figure 2. Compound **10** underwent biocatalytic reductive amination using the wild-type *Sp*RedAm enzyme from *S. purpureus*, which gave the *cis*-amino ester compound **11** in 74% yield. The reaction of **11** and **12** afforded the isopropyl ester **13**. The reaction of **13** with hydroxylamine hydrochloride afforded the hydroxamic acid derivative **14**, which underwent Lossen rearrangement on the reaction with 1,1-carbonyldiimidazole (CDI) to give **15**. The sulfonylation of **15** was achieved using the triazole derivative **16** to avoid the sulfonylation of the pyrrole nitrogen.

##### Target Kinases

The inhibitory activity of abrocitinib against JAK kinases was evaluated by Vazquez et al. [59]. The results revealed inhibitory activity against JAK1 and JAK2 at IC_50_ values of 0.029 and 0.803 μM, respectively (Figure 12).

##### Crystal Structures

Abrocitinib exists as a co-crystallized ligand in two crystal structures, including its crystal structure with JAK1 (pdb: 6BBU) and JAK2 (pdb: 6BBV) [59].

The binding mode of abrocitinib in the active site of JAK1 is illustrated in Figure 13. Abrocitinib shows three conventional hydrogen bonds with Glu957, Leu959, and Asn1008. In addition, abrocitinib shows three carbon–hydrogen bonds with LEU881 and Leu959. Several hydrophobic interactions were also observed between abrocitinib and amino acids in JAK1.

On the other hand, abrocitinib exhibited four conventional hydrogen bonding interactions with Glu930, Leu932, Arg980, and Asn981, Figure 14. The binding interactions of abrocitinib also included one carbon–hydrogen bond with Leu932 and several hydrophobic interactions, with Leu855, Val863, Ala880, Val911, and Leu983.

##### Pharmacological Activities and Uses

Abrocitinib combined with topical therapy displayed higher effectiveness in the treatment of atopic dermatitis compared to the placebo [73]. In another clinical trial (NCT03720470), abrocitinib showed higher therapeutic benefits regarding the symptoms of atopic dermatitis compared to the placebo [74].

The abrocitinib-induced response in patients with moderate-to-severe atopic dermatitis was also evaluated by Blauvelt et al. [75]. The results revealed the effectiveness of the induction treatment with abrocitinib, as most responders did not flare.

##### Metabolism

The metabolic study of abrocitinib in humans revealed the formation of several oxidative metabolites, in addition to the parent drug (26%) [76,77]. The results of the in vitro metabolic study of abrocitinib revealed that it undergoes metabolism by several CYP450 enzymes, where CYP2C19 and CYP2C9 were the major metabolizing enzymes. Among these metabolites, the 3-hydroxypropyl M1 (PF-06471658) and the 2-hydroxypropyl metabolite M2 (PF-07055087) were retrieved in 11% and 12%, respectively, Figure 15. Both M1 and M2 exhibited JAK inhibitory activity similar to the parent drug with higher selectivity toward JAK1 over the other JAKs. On the other hand, a third oxidative metabolite M4 (PF-07054874) that lacks JAK inhibitory activity was also isolated [77].

#### 1.7.2. Baricitinib

##### Approval History

Baricitinib (Figure 16) is an orally active small-molecule inhibitor of JAK1/2. It was approved by the European Medicine Agency (EMA) in 2017 for the treatment of rheumatoid arthritis [65]. In June, 2018, baricitinib was approved by the FDA for the treatment of moderate-to-severe rheumatoid arthritis in adults [64]. Recently, the FDA issued an emergency use authorization for the combination of baricitinib and remdesivir to treat hospitalized patients with COVID-19 [78].

##### Synthesis

The original synthesis of baricitinib was reported by Rodgers et al. [79]. In the first step, 3-oxoazetidine-1-carboxylate **18** underwent a Horner–Emmons reaction with diethyl (cyanomethyl)phosphonate **17** to give **19**, Figure 3. Treatment of **19** with hydrochloric acid resulted in the removal of the protective group and produced **20**. The reaction of compound **20** with ethane sulfonyl chloride **21** afforded **22**.

On the other hand, protection of the pyrrole nitrogen in **12** was achieved on the reaction of 12 with (2-(chloromethoxy)ethyl)trimethylsilane (SEM) 23, which gave **24**. Compound **24** underwent Suzuki coupling with 4-(1,3,2-dioxaborolan-2-yl)-1*H*-pyrazole **25** to give **26**, Figure 3. The acid hydrolysis of **26** afforded **27**, which was then reacted with **22** to give compound **28**. Compound **28** was treated with lithium tetrafluoroborate followed by ammonia solution to give baricitinib.

Xu et al. [80] have reported an efficient five-step synthetic route of baricitinib (Figure 4) with an overall yield of 49% compared to the eight-step method reported by Rodgers et al. [79]. Both methods share the first 3 steps. In the fourth step of Xu et al.’s method, a nucleophilic addition of **29** to the double bond in **22** afforded compound **30**. Suzuki coupling of **30** with compound **12** gave baricitinib. This method may be suitable for industrial synthesis due to its low cost and simple operating requirements [80].

Furthermore, several other methods have been reported, describing alternative synthetic routes of baricitinib using the key intermediates **12** and **22** [81,82]. Among these methods, Cui et al. [82] reported a green and facile synthesis of the azetidine intermediate **22** using inexpensive and environmentally friendly starting materials.

##### Target Kinases

The inhibitory activity of baricitinib against JAKs was evaluated by Clark et al. [83]. These results revealed high inhibitory activity against JAK1/2 (Figure 17). However, baricitinib also exhibited moderate inhibitory activity against TYK2 (IC_50_ = 61 nM).

##### Crystal Structures

Baricitinib is found in three crystal structures in the protein data bank. The first is the crystal structure of baricitinib with BMP-2-inducible kinase (pdb: 4W9X) [84]. In addition baricitinib also exists as a co-crystallized ligand in two crystal structures of JAK2 [85,86]. 

The bone morphogenic proteins (BMPs) are involved in skeletal morphogenesis [87]. The binding mode of baricitinib int BMP-2-inducible kinase is visualized in Figure 18. The binding interactions of baricitinib into BMP-2-inducible kinase include 3 conventional hydrogen bonds with Ser63, Cys133 and Gln137. Baricitinib also shows one carbon hydrogen bond with Asp198 and one pi-sulfur interaction with Met130. Several hydrophobic interactions could also be observed between baricitinib and the hydrophobic residues in the kinase.

Visualization of the binding mode/interactions of baricitinib into JAK2 JH1 (pdb: 6VN8) shows one conventional hydrogen bond with Leu932 [85]. Baricitinib also shows three carbon hydrogen bonds with Ser862, Lys882, and Asp994 (Figure 19). Baricitinib also shows one pi-sulfur interaction with Met929 and several hydrophobic interactions of the pi-sigma and pi-alkyl types.

In addition, the binding mode of baricitinib into human JAK2 JH1 was also reported in another crystal structure (pdb: 6WTO) [86]. Visualization of the binding mode/interactions (Figure 20) shows identical hydrogen bonds and hydrophobic interactions with those of the above crystal (pdb: 6VN8), Figure 19.

##### Pharmacological Activities and Uses

Baricitinib was the second JAK inhibitor to be approved for rheumatoid arthritis after the approval of ruxolitinib. It showed rapid and long-lasting therapeutic benefits in the treatment of rheumatoid arthritis [88].

Baricitinib also improved the signs and symptoms of systemic lupus erythematosus in a double-blind study [89]. Moreover, baricitinib decreased inflammation/pruritus in atopic dermatitis patients when used with topical corticosteroids [90].

On the other hand, The inhibition of JAK1/2 by baricitinib leads to the inhibition of the JAK-STAT signaling pathway and subsequent inhibition of the production of the pro-inflammatory cytokines that could be useful for patients with COVID-19 [91]. The results of the clinical trial of baricitinib in combination with remdesivir (NCT04401579) revealed superior activity in reducing the recovery period compared to remdesivir alone [92]. Similar results were also obtained when using a combination of dexamethasone with baricitinib/remdesivir [93]. Finally, the combination of baricitinib and remdesivir received an Emergency Use Authorization (EUA) from the FDA to treat hospitalized patients with COVID-19.

##### Metabolism

The in vivo metabolism of baricitinib was performed in mice, rats, and dogs using [^14^C]-baricitinib [94,95]. The metabolic pathways include mono- and bis-oxidation, oxidative ring opening, and oxidative decarboxylation. The in vitro metabolism of baricitinib incubated with human liver enzymes was also very limited, with the formation of four metabolites.

Baricitinib undergoes minor in vivo metabolic transformation mediated by the CYP3A4 enzyme [94,95]. Nearly 6–10% of baricitinib undergoes metabolism, producing four oxidative metabolites (Figure 21). Accordingly, it is excreted mainly as an unchanged drug in urine (69%) and feces (15%), while no circulating metabolites were identified in human plasma. The proposed metabolite M3 was formed through oxidative ring opening of the pyrrole ring, which undergoes oxidative decarboxylation giving M12. On the other hand, mono- or bis-oxidation of baricitinib affords the oxidative metabolites M10 and M22, respectively. The metabolic profile of baricitinib in animals was similar to that observed in humans.

#### 1.7.3. Delgocitinib

##### Approval History

Delgocitinib (Figure 22) is a pan-JAK inhibitor that was approved for atopic dermatitis in 2020 in Japan [69]. Delgocitinib demonstrated inhibitory activity against the four JAKs [96].

##### Synthesis

Noji et al. have reported the original stereoselective synthesis of delgocitinib [96]. In the first step, compound **31** was alkylated with *t*-butyl bromoacetate to give **32**, Figure 5. The chlorination of **32** resulted in the formation of the reactive aziridinium intermediate **34**, which gave **35** on heating. Compound **35** underwent intramolecular cyclization to afford **36** as a single enantiomer. The protecting group in **36** was replaced by *t*-butyloxycarbonyl protecting (Boc) in **37**. Compound **37** underwent *α*-alkylation to give **38**, which underwent ozonolysis and reductive amination to give **39**. Removal of the Boc group and intramolecular cyclization of **39** afforded **40**, which was reacted with di-*tert*-butyl dicarbonate to protect the azetidine nitrogen, resulting in **41**. The debenzylation of **41** followed by reaction with 4-chloro-7*H*-pyrrolo[2,3-*d*]pyrimidine **12** afforded **43**, which underwent deprotection by the removal of the Boc group, followed by cyanoacetylation, using **45** to give delgocitinib.

In addition, Takiguchi et al. have also reported the stereocontrolled synthesis of delgocitinib [97]. The synthesis was started with 3-bromodihydrofuran-2(3*H*)-one **46**, which was converted to **49** in a three-step synthesis, Figure 6. Compound **49** underwent intramolecular cyclization catalyzed by Cs_2_CO_3_ to afford a diastereomixture of **50** and **51** in 98% and 2% yield, respectively. Compound **50** was isolated chromatographically and reacted with potassium phthalimide, followed by esterification with ethyl iodide to give the intermediate **52**. Dephthaloylation of compound **52** gave the dilactam derivative **53**. The LiAlH4/TMSCl-catalyzed reduction of **53** afforded **54**, which was isolated as a stereochemically pure succinate salt in 86% yield. The reaction of **12** with **54** afforded compound **44**, which was reacted with the cyanoacetyl pyrazole derivative **45** to give delgocitinib.

##### Target Kinases

Noji et al. [96] have investigated the kinase inhibitory activity of delgocitinib. The results (Figure 23) show potent activities against the four JAKs (IC_50_ = 2.6–58 nM). Delgocitinib also exhibited inhibitory activity against lymphocyte-specific protein tyrosine kinase (LCK) at IC_50_ value of 5.8 μM.

##### Crystal Structures

One crystal structure of delgocitinib in complex with JAK3 (pdb: 7C3N) is available on the protein data bank [96]. The binding mode of delgocitinib into JAK3 is visualized in Figure 24. Delgocitinib shows one conventional hydrogen bond with Leu905 and an electrostatic interaction (attractive charge) with Asp967.

##### Pharmacological Activities and Uses

Delgocitinib displayed therapeutic efficacy in the treatment of patients with atopic dermatitis [98]. Delgocitinib was approved in Japan for atopic dermatitis [69].

Due to its pan-JAK inhibitory activity, the pharmacological effects of delgocitinib depend on the competitive inhibition of the four JAKs, which plays an important role in the pathophysiology of chronic inflammatory skin diseases [69]. Delgocitinib also inhibited the activation of inflammatory cells such as mast cells and T cells [99].

Delgocitinib was also evaluated for the treatment of psoriasis and chronic hand eczema (CHE) [69]. After 8 weeks of topical treatment, delgocitinib showed clearance of CHE in a significant number of patients compared to the vehicle [100]. On the other hand, the clinical trial of delgocitinib for discoid lupus erythematosus (NCT03958955) was terminated [101].

##### Metabolism

The pharmacokinetic study of delgocitinib (JTE-052) revealed that it is excreted mainly as the parent drug in urine, which indicates its stability to metabolism [102]. Delgocitinib also has higher stability values in human cells compared to animal cells [96,103]. The in vitro metabolic study performed using mouse, rat, rabbit, and dog hepatocytes revealed the recovery of the parent drug in 95.5–97.7%. These results indicate that less than 5% of the total dose was metabolized. The metabolic study was performed using animal microsomes and revealed the formation of three metabolites, M1 (oxidative metabolites of the side chain), M2, and M3 (oxidative metabolites of pyrrolopyrimidine), Figure 25.

#### 1.7.4. Fedratinib

##### Approval History

Fedratinib (Figure 26) is a competitive inhibitor of JAK2, BRD4, and FMS-like tyrosine kinase 3 (FLT3) [35]. Fedratinib was approved by the FDA in August 2019 to treat patients with intermediate-2 or high-risk primary or secondary myelofibrosis [66].

##### Synthesis

The synthesis of fedratinib could be achieved by the reaction of the intermediates **58** and **61** [104]. Firstly, compound **58** was obtained from the reaction of *p*-nitrophenol **55** and 1-(2-chloroethyl)pyrrolidine **56** in the presence of CsCO_3_ according to the previous report [105]. The reduction of the nitro group in **57** afforded the amino ether **58**, Figure 7. 

On the other hand, compound **61** was obtained in 98% yield from the Pd-catalyzed coupling of **59** and **60** [104,106,107]. Compound **61** could also be prepared in 79% yield from the reaction of **62** with **63** in methanol. The reaction of **58** and **61** in the presence of sodium hydroxide followed by treatment with acidic isopropyl alcohol gave fedratinib, Figure 7.

##### Target Kinases

Wernig et al. investigated the kinase inhibitory activity of fedratinib [35]. The results revealed the highest inhibitory activity against JAK2 and JAK2V617F at IC_50_ 3 nM against the two kinases, Figure 27. On the other hand, fedratinib exhibited weak inhibitory activity against JAK3. Fedratinib also showed inhibitory activity against FLT3 and RET kinases with IC_50_ values of 15 nM and 48 nM, respectively. In addition, fedratinib exhibited potent bromodomain inhibitory activity at IC_50_ value in the nanomolar range [108].

##### Crystal Structures

Three crystal structures of fedratinib are available on the protein data bank. Two of these crystals are bromodomain of human dual kinase-bromodomain (BRD4) in complex with fedratinib (pdb: 4OGJ and 4PS5) [108,109], while the third crystal is a complex of fedratinib with JAK2 JH1 (pdb: 6VNE) [85].

The binding mode of fedratinib into human bromodomain BRD4 was visualized in Figure 28. Fedratinib shows two conventional hydrogen bonds with Asn140 and multiple hydrophobic interactions with BRD4.

On the other hand, fedratinib displays two conventional hydrogen bond and Asn140 into human bromodomain BRD4 (pdb: 4PS5). However, fedratinib also shows an additional hydrogen bond with Pro82 (Figure 29).

In addition, the binding mode of fedratinib into JAK2 JH1 was visualized in Figure 30. Fedratinib exhibits two conventional hydrogen bonds with Leu932 in JAK2 JH1. Moreover, two carbon hydrogen bonds could be observed with Leu855 and Glu930.

##### Pharmacological Activities and Uses

Fedratinib has equal inhibitory activity against the wild-type JAK2 and its mutated form, JAK2V617F [110]. It was approved for the treatment of myelofibrosis in adult patients [66].

Moreover, fedratinib displayed inhibitory activity against other kinases such as FLT3 and RET kinases (Figure 27). The overexpression of FLT3 has been reported in acute leukaemia [111]. In addition, the overexpression of BRD proteins has been reported in different types of cancers [112]. Accordingly, the inhibition of BRD4 by fedratinib could contribute to its pharmacological activity [110].

##### Metabolism

Ogasawara et al. evaluated the metabolic transformation of [^14^C]-labeled fedratinib in healthy subjects [113]. The results revealed that nearly 80% of the radioactivity in plasma was due to the parent compound. Among the identified metabolites, SAR317981, a metabolite obtained through oxidation of the pyrrolidine ring, was retrieved in 9.4%, Figure 31. In addition, the *N*-butyric acid derivative (SAR318031) was retrieved at 5.7%. Fedratinib and its metabolites are eliminated mainly through feces (50–70%) and partly in urine (~3%). The amino ethanoic acid derivatives (M17 and M21b) were also detected among 19 different metabolites identified in feces.

#### 1.7.5. Filgotinib

##### Approval History

Filgotinib (Figure 32) is classified as a selective, ATP-competitive inhibitor of JAK1 [70]. Filgotinib was approved by the European Medicines Agency (EMA) in September 2020, for adult patients with moderate to severely active rheumatoid arthritis [70]. Filgotinib was also approved for the treatment of rheumatoid arthritis in Japan [70].

##### Synthesis

The synthesis of filgotinib (Figure 8) was achieved in a five-step synthesis [114,115]. Initially, compound **64** was reacted with thiomorpholine 1,1-dioxide **65** to give **66**. On the other hand, the triazolopyridine **70** was prepared from compound **67** in a two-step synthesis. The first step involved the reaction of **67** with ethoxycarbonyl isothiocyanate **68** to give **69**, which underwent intramolecular cyclization with the hydrolysis of the carbamate group to give compound **70**. Acylation of the amino group in **70** with cyclopropanecarbonyl chloride **71** afforded **72**. Finally, filgotinib was obtained from the reaction between **66** and **72**.

##### Target Kinases

The inhibitory activity of filgotinib was evaluated using recombinant JAKs and whole blood assay [116,117]. The results revealed the highest inhibitory activity against JAK1 (IC_50_ = 10 nM), Figure 33. In addition, filgotinib exhibited 2.8-, 81-, and 116-fold selectivity for JAK1 over JAK2, JAK3, and TYK2, respectively.

Filgotinib was also tested against a panel of 170 kinases [117]. The results revealed weak inhibitory activities against hFLT3, hFLT4 and hCSF1R (Figure 33).

##### Crystal Structures

Filgotinib exists in two crystal structures with JAKs protein in the protein data bank. The first crystal (pdb: 4P7E) [117] was refined at a resolution of 2.40 Å, while the second (pdb: 5UT5) [118] was refined at 1.90 Å resolution. The binding mode/interactions of filgotinib into the ATP-binding site of JAK1 (pdb: 4P7E) shows two conventional hydrogen bonds with Leu932 and one carbon hydrogen bond Lys882, Figure 34. 

On the other hand, the binding mode of filgotinib into the JH2 domain of JAK2 was visualized in Figure 35. Filgotinib shows two conventional hydrogen bonds with Val629 and two carbon hydrogen bonds with Gln553 and Ser633.

##### Pharmacological Activities and Uses

Filgotinib was first investigated for the treatment of rheumatoid arthritis, where the clinical results proved that rheumatoid arthritis can be treated with a selective inhibitor of JAK1 [119]. Later, filgotinib showed promising efficacy in the treatment of rheumatoid arthritis in two randomized phase IIa trials of filgotinib that were performed by Vanhoutte et al. [120]. In addition, filgotinib showed a rapid improvement in the signs and symptoms of rheumatoid arthritis [121]. In 2020, filgotinib was approved for the treatment of moderate-to-severe rheumatoid arthritis. It acts as a selective inhibitor of JAK1, which leads to the prevention of STAT phosphorylation and activation [3].

Filgotinib was also evaluated in a phase-II study for the treatment of Crohn’s disease [122]. The results obtained from this study were also promising for patients with moderate-to-severe Crohn’s disease. Moreover, the clinical remission induced by filgotinib in patients with active Crohn’s disease was associated with an acceptable safety profile [123]. 

Moreover, the efficacy of filgotinib in the treatment of ulcerative colitis was also evaluated in a clinical trial (NCT02914522) [124]. The results revealed the efficacy of filgotinib in the treatment of ulcerative colitis.

In addition, the safety and efficacy of filgotinib were also evaluated in a clinical trial (NCT03101670) for the treatment of active psoriatic arthritis [125].

##### Metabolism

Namour et al. [126] investigated the pharmacokinetics of filgotinib and its active metabolite in healthy male volunteers. The results revealed the formation of an active metabolite, aminotriazolopyridine metabolite (Figure 36), which contributes to the inhibitory activity against JAKs [119,127]. In addition, the active metabolite of filgotinib undergoes further metabolism by glucuronidation of the free amino group to give a glucuronide metabolite.

The aminotriazolopyridine active metabolite of filgotinib (Figure 36) exhibited an inhibitory activity against human JAK1 (IC_50_ = 307 nM) using human recombinant JAK1 [117,127]. On the other hand, the inhibitory activity of this metabolite against JAK1 was observed at IC_50_ of 11.9 μM in a whole blood assay (IL6-induced pSTAT1).

In another study, Namour et al. identified the aminotriazolopyridine active metabolite of filgotinib in three animals (mouse, monkey, and dog) and humans [128]. The study revealed a significant interaction between filgotinib or its active metabolite with CYP450, glucuronosyltransferases, or drug transporters. This study suggests that filgotinib can be combined with the other drugs used for rheumatoid arthritis.

#### 1.7.6. Oclacitinib

##### Approval History

In 2013, oclacitinib (Figure 37) was approved by the FDA to treat dogs with atopic dermatitis and pruritus associated with allergic dermatitis [129].

##### Synthesis

The preparation of oclacitinib was patented by Berlinski et al. [130]. The synthetic route depends on the reaction of the commercially available 4-chloro-7*H*-pyrrolo[2,3-*d*]pyrimidine **12** with tosyl chloride to give compound **6**, Figure 9.

In addition, compound **6** was reacted with **73** to afford **74**, which underwent sulfonylation with methanesulfonyl chloride **75** to give **76**, Figure 9. The reaction of **76** with sodium sulfite gave **77**, which was treated with thionyl chloride to give **78**. Finally, the reaction of compound **78** with methylamine gave **79**, which was treated with lithium hydroxide to liberate oclacitinib as a free base.

##### Target Kinases

Gonzales et al. evaluated the inhibitory activity of oclacitinib against the four JAKs using isolated enzymes [129]. The results revealed inhibitory activity against the four kinases at IC_50_ values in the range of 10–99 nM, Figure 38. The highest inhibitory activity of oclacitinib was observed against JAK1 (IC_50_ = 10 nM) with 1.8- and 9.9-fold selectivity toward JAK1 compared to JAK2 and JAK3, respectively.

Oclacitinib was also evaluated for its inhibitory activity against a panel of 38 non-JAK kinases [129]. The results revealed 32%–42% inhibition in the activity of aurora-related kinase 1 (ARK1), serine/threonine protein kinase MARK, high-affinity nerve growth factor receptor (TRK-A), and vascular endothelial growth factor receptor 2 (VEGFR-2/FLK1), Figure 38.

##### Crystal Structures

Oclacitinib has not yet been reported in a crystal structure with any of its target kinases.

##### Pharmacological Activities and Uses

Gonzales et al. evaluated the effect of oclacitinib on the function of different cytokines in canine and human cell model systems [129]. The results revealed the inhibition of IL-2, IL-4, IL-6, IL-13, and IL-31, which play an essential role in inflammation, allergy, and pruritus at IC_50_ values in the range of 36–249 nm. In 2013, oclacitinib was approved by the FDA to treat dogs ≥12 months old with atopic dermatitis and pruritus associated with allergic dermatitis [129]. Moreover, Haugh et al. also reported the first use of oral oclacitinib in the treatment of a man with atopic dermatitis [131].

Oclacitinib also showed safe and effective control in the treatment of dogs with pruritus associated with allergic dermatitis [132].

Rynhoud et al. investigated the association between the use of oclacitinib and antibacterial therapy [133]. The results suggested a reduction of the antibacterial use when combined with oclacitinib, compared to other antipruritic agents.

Banovic et al. [134] have reported an immunosuppressive effect of oclacitinib in dogs when used at a dose higher than that used in the treatment of allergic pruritus. However, an increase in CD4+ lymphocyte populations in dogs was observed after long-term treatment with oclacitinib [135].

##### Metabolism

Oclacitinib maleate has rapid oral absorption with 89% absolute bioavailability and a low clearance rate in dogs [136]. It undergoes metabolism into several metabolites. Among these metabolites, one major oxidative metabolite was detected in plasma and urine [137]. However, oclacitinib exhibited very weak inhibitory activity against Cyp450 enzymes, which indicates a low potential for drug–drug interaction.

#### 1.7.7. Pacritinib

##### Approval History

Pacritinib (Figure 39), a JAK2/FLT3 inhibitor, was approved by the FDA for the treatment of myelofibrosis in adult patients with thrombocytopenia [138].

##### Synthesis

The synthesis of pacritinib was reported by William et al. [139]. It depends on the reaction of the intermediate compounds **83** and **88**, Figure 10.

At the first step (Figure 10), compound **80** was reacted with **81** to give **82**, which underwent a base-catalyzed alkylation using allyl bromide to give **83** [139].

On the other hand, 2-hydroxy-5-nitrobenzaldehyde **84** was reacted with dichloroethane, followed by a reduction in the resulting product **85** to give the corresponding alcohol **86** [139]. Compound **86** was alkylated using allyl bromide to give **87**, which saw a decrease in the nitro group to give **87**. The reaction of **88** with **83** afforded **89**, which was reacted with pyrrole to give pacritinib, Figure 10.

##### Target Kinases

The kinase inhibitory activity of pacritinib was evaluated by William et al. [139]. The results revealed inhibitory activity against JAK2 and JAK2^V617F^, at IC_50_ values of 22 and 19 nM, respectively, Figure 40.

Pacritinib also showed weaker inhibitory activity against the other types of JAK kinases. It also showed potent inhibitory activity against FLT3 (IC_50_ = 22 nM) [139]. In addition, only weak inhibitory activity was observed against CDK2, Figure 40.

Pacritinib also underwent extensive evaluation of its inhibitory activity against 439 recombinant kinases [140]. Besides the inhibition of JAK2, JAK2V617F, and FLT3, pacritinib also showed inhibitory activity against colony-stimulating factor 1 receptor, and interleukin-1 receptor-associated kinase 1 at IC_50_ < 50 nM.

##### Crystal Structures

Pacritinib exists in only one crystal structure in the protein data bank. This crystal includes pacritinib in complex with human quinone reductase 2 (NQO2) (pdb: 5LBZ) [141]. Pacritinib shows only multiple hydrophobic interactions of the pi-pi stacked and pi-pi T-shaped types with Phe126 and Phe178 in this reductase enzyme, Figure 41. However, no crystal structure with any of the target JAKs has been reported yet.

##### Pharmacological Activities and Uses

Pacritinib acts as an inhibitor of both JAK2 and FLT3 that could be used to overcome the resistance problems in patients with acute myeloid leukaemia (AML) [30]. The dual activity against the two kinases could enable pacritinib to overcome the resistance mediated by the upregulation of JAK2 in FLT3-TKI-resistant AML cells [30].

Pacritinib was later tested for its activity in the treatment of myelofibrosis in several clinical trials [142,143]. The efficacy of pacritinib was evaluated against the best available therapy in patients with myelofibrosis (NCT01773187) [142]. The results revealed a significant reduction in the volume of the spleen (SVR) in patients receiving pacritinib therapy, indicating that it could be used in the treatment of myelofibrosis.

In addition, pacritinib also showed higher efficacy in reducing splenomegaly than the best available therapy in patients with myelofibrosis and thrombocytopenia (NCT02055781) [143].

Pacritinib also showed potential anti-leukemic activity when combined with chemotherapeutic agents [144,145].

The efficacy of pacritinib in the treatment of patients with glioblastoma multiforme was also reported when combined with temozolomide [146].

##### Metabolism

The preliminary results of the metabolic study of pacritinib revealed that CYP3A4 is the main metabolizing enzyme [139]. However, no inhibitory activity was observed against the other isoenzymes.

On the other hand, the metabolism of pacritinib was investigated in vitro/in vivo by Jayaraman et al. [147]. The results revealed the formation of four metabolites by liver microsomes in both humans and mice, Figure 42.

The detected metabolites include two oxidation metabolites formed by oxidation of the pyrrole ring (M1) and pyrrole nitrogen (M3) [147]. In addition, the third metabolite (M2) formed by *O*-dealkylation of the pyrrole-bearing side chain, while the fourth one (M4) was formed by reduction of the double bond, Figure 42.

#### 1.7.8. Peficitinib

##### Approval History

Peficitinib (Figure 43) is a pan-JAK inhibitor. It was approved in 2019 in Japan for the treatment of rheumatoid arthritis [68].

##### Synthesis

Peficitinib was obtained from pyrrolo[2,3-*b*]pyridine **7** in a seven-step synthesis [104,148]. In the first step, *N*-protected pyrrolo[2,3-*b*]pyridine **90** was obtained from the reaction of **12** with triisopropylsilyl chloride (TIPSCl). Compound **91** was obtained from the reaction of **90** with *sec*-BuLi and ethyl chloroformate followed by treatment with *tetra*-butylammonium fluoride (TBAF) to remove the TIPS group, Figure 11.

Conversion of the ethyl ester group in **91** to the corresponding amide group in **92** was achieved in two steps (Figure 11). The first step proceeded through hydrolysis of the ester group in **91** to liberate the corresponding carboxylic acid derivative, which was reacted with ammonium hydroxide in the presence of CDI to give the amide derivative **92**.

On the other hand, the diastereomeric mixture of **93** was reacted with benzyl chloroformate followed by chromatographic separation to isolate the *trans*-isomer **95**. Pd/C-catalyzed hydrogenation of **95** yielded *trans*-4-aminoadamantan-1-ol **96**. The reaction between **92** and **96** afforded peficitinib as a free base. Treatment of peficitinib with hydrobromic acid afforded peficitinib hydrobromic acid salt, Figure 11.

##### Target Kinases

The kinase inhibitory activity of peficitinib was evaluated against the four JAKs [149,150]. The results (Figure 44) revealed inhibitory activity against the four kinases with IC_50_ values in the range of 0.70–5.0 nM [149]. These results indicate that the highest inhibitory activity of peficitinib was against JAK3.

##### Crystal Structures

Peficitinib is available as a co-crystallized ligand in four crystal structures with JAKs in the protein data bank. These crystals include the crystal structure of peficitinib bound to JAK1 (pdb: 6AAH) [151], JAK2 (pdb: 6AAJ) [151], JAK3 (pdb: 6AAK) [151], and TYK2 (6AAM) [151]. All at 1.83–2.67 Å resolution. The binding mode and interactions of peficitinib with JAK1 (pdb: 6AAH) are presented in Figure 45. Peficitinib shows three conventional hydrogen bonds with Leu959 and Asn1008.

The binding interactions of peficitinib in JAK2 (pdb: 6AAJ) show two conventional hydrogen bonds with Glu930 and Leu932, Figure 46. In addition, peficitinib forms one carbon hydrogen bond with Glu930.

The binding mode and interactions of peficitinib are presented in Figure 47. Peficitinib shows two conventional hydrogen bonds with Glu903 and Leu905, as well as one carbon–hydrogen bond with Glu903.

On the other hand, the binding orientation, and interactions of peficitinib with TYK2 are presented in Figure 48. Peficitinib shows four conventional hydrogen bonds: Val981, Glu979, Asn1028, and Asp1041.

##### Pharmacological Activities and Uses

Cytokines play an important role in pain, inflammatory reactions, and nerve sensitization [152]. The pan-JAK inhibitor, peficitinib, was evaluated for the treatment of rheumatoid arthritis, which is characterized by joint destruction and inflammation [153]. Peficitinib suppressed bone destruction and paw swelling in rats with adjuvant-induced arthritis [149].

In a clinical trial (NCT01565655), peficitinib exhibited a dose-dependent ACR20 response rate when given orally to patients with moderate-to-severe rheumatoid arthritis [154]. Peficitinib also demonstrated clinical efficacy and prevention of joint destruction in Asian patients who displayed an inadequate response to conventional DMARDs [155]. In addition, no drop in the effectiveness of peficitinib was observed after long-term use [156]. However, the use of peficitinib was also associated with a risk of herpes zoster infection, similar to other JAK inhibitors [157].

##### Metabolism

Oda et al. investigated the metabolism of peficitinib in healthy male subjects using [^14^C]-labelled peficitinib [158]. The results revealed the formation of a sulfate-conjugated metabolite (M2). The sulfate conjugate and the parent drug constituted the major components in plasma and urine, Figure 49. On the other hand, peficitinib was metabolized to give the *N*-methylated derivative (M4), which was identified in feces.

Miyatake et al. investigated the effect of hepatic impairment on the pharmacokinetic profile of peficitinib [159]. The results of this study suggest a decrease in the dose of peficitinib in patients with moderate hepatic impairment.

#### 1.7.9. Ruxolitinib

##### Approval History

Ruxolitinib (Figure 50) was approved by the FDA in November 2011 to treat myelofibrosis [28]. The FDA also approved ruxolitinib in December 2014 for the treatment of patients with polycythemia vera [29]. In addition, ruxolitinib received approval in 2019 and 2021 for the treatment of acute and chronic graft-versus-host disease, respectively [160].

##### Synthesis

The synthesis of ruxolitinib (Figure 12) was achieved from compound **12** in a five-step synthesis [115,161]. Compound **12** was first reacted with 2-(trimethylsilyl)ethoxyethyl chloride (SEM-Cl) to give **24**.

In addition, compound **27** was prepared from compound **24** via two steps, Figure 12. The first step is a Suzuki coupling of **24** and **25** to give **26**. In the second step, compound **26** underwent acid-catalyzed hydrolysis to give **27**. The reaction of **27** with 3-cyclopentylacrylonitrile **97** afforded a mixture of *R*- and *S*-enantiomers **98**, which underwent chiral separation. The *R*-enantiomer of **98** was treated with trifluoroacetic acid to give ruxolitinib.

Haydl et al. [162] reported a regio- and enantioselective synthesis of *N*-substituted pyrazoles that can be used in the synthesis of ruxolitinib, Figure 13. Compound **101** was prepared from the reaction of cyclopentyl magnesium bromide **99** and 3-bromoprop-1-yne **100**.

Asymmetric addition reaction of the substituted pyrazole **102** to the cyclohexylallene **101** afforded **103**; Figure 13. Hydroboration of **103** using 9-borabicyclo(3.3.1)nonane followed by the Swern oxidation of the alcoholic group afforded **105**. Compound **105** was then reacted with hydroxylamine in the presence of iodine to give the corresponding nitrile **106**, which was then reacted with bis(pinacolato)diboron to give **107**. The Suzuki coupling of compounds **12** and **107** afforded (*R*)-ruxolitinib.

Lin et al. [163] have also reported an enantioselective synthesis of ruxolitinib (INCB018424). This synthesis depends on the addition of a substituted pyrazole to (*E*)-3-cyclopentylacrylaldehyde, which was catalyzed by diarylprolinol silyl ether.

##### Target Kinases

Clark et al. investigated the inhibitory activity of ruxolitinib against JAKs [83]. The results (Figure 51) revealed the highest inhibitory activity against JAK1 and JAK2 with IC_50_ values of 6.4 and 8.8 nM, respectively.

The kinase inhibitory activity of ruxolitinib was also evaluated against 368 kinases by Zhou et al. [164]. At 1 μM, ruxolitinib displayed 96%-100% inhibition in the activity of the four JAKs. Ruxolitinib also showed inhibitory activity against other kinases such as FAC, RET, TRK-B, and LRRK2 (wild-type), Figure 51. Ruxolitinib also exhibited inhibitory activity against c-Src kinase at IC_50_ value of 2.92 nM [165].

In another study performed by Quintás-Cardama et al. [166], ruxolitinib displayed inhibitory activity against the four JAKs at IC_50_ in the range of 2.8–428 nM. The results of this study revealed the inhibitory activity of ruxolitinib against JAK1 (IC_50_ = 3.3 nM) and JAK2 (IC_50_ = 2.8 nM). Ruxolitinib also exhibited moderate inhibitory activity against TYK2 (IC_50_ = 19 nM). In addition, high efficacy of ruxolitinib was also observed in tumour cells with JAK2V617F mutation.

Ruxolitinib was also evaluated for its inhibitory activity against CHK2 and c-Met kinases, where the results revealed IC_50_ values exceeding 1000 nM [166].

##### Crystal Structures

Ruxolitinib exists as a co-crystallized ligand in five crystal structures in the protein data bank. Three of these crystals include complexes of ruxolitinib with the JH1 domain of JAK2 (pdb: 6VGL, 6WTN, and 6VNK) [85,86]. Ruxolitinib also exists as a co-crystallized ligand with c-Src (pdb: 4U5J) [165] and DRLK1 (pdb: 7F3G). The binding mode and interactions of ruxolitinib into JAK2 JH1 (pdb: 6VGL) are visualized in Figure 52. Ruxolitinib shows one conventional hydrogen bond with Leu932 and two carbon hydrogen bonds with Leu855 and Gly856.

Ruxolitinib also exists as a co-crystallized ligand into the JH1 domain of human JAK2 (pdb: 6WTN) at 1.83 Å resolution. Visualization of the binding mode and interactions of ruxolitinib shows one conventional and one carbon hydrogen bond with Leu932, Figure 53.

In addition, ruxolitinib was also isolated as a co-crystallized ligand with JAK2 JH1 (pdb: 6VNK) at 2.00 Å resolution [85]. The binding interactions also shows one conventional hydrogen bond with Leu932 and one carbon hydrogen bond with Gly856, Figure 54.

The Src and JAK family kinases share ~34% sequence identity in the kinase domain [165]. Ruxolitinib was also refined in a crystal structure with c-Src (pdb: 4U5J) [165]. The binding interactions of ruxolitinib with c-Src show one conventional hydrogen bond with Met341 and one carbon–hydrogen bond with Glu339, Figure 55.

Ruxolitinib also exists as a co-crystallized ligand with DCLK1 kinase (pdb: 7F3G). Visualization of the binding mode/interactions of ruxolitinib into DCLK1 shows one conventional hydrogen bond with Val468, Figure 56.

##### Pharmacological Activities and Uses

Ruxolitinib is a JAK1/2 inhibitor which showed the strong inhibition of JAK2V617F-positive Ba/F3 cells [167]. The efficacy of ruxolitinib in the treatment of myelofibrosis was evaluated in several clinical trials [167,168]. The results of the clinical trial (NCT00952289) revealed significant therapeutic outcomes compared to placebo [168].

In addition to its inhibitory activity against JAKs, ruxolitinib also targets other kinases such as CHK2 and c-Met [166]. Furthermore, ruxolitinib exhibited antiproliferative activity against JAK2V617F+ Ba/F3 cells at IC_50_ 127 nM [166]. The combination of ruxolitinib and ERBB1/2/4 inhibitors also displayed synergistic anticancer activity against lung, breast, and ovarian cancer cells [169].

In 2019, Kim et al. [170] investigated the efficacy of ruxolitinib cream in the treatment of atopic dermatitis in a phase 2 study. A fast improvement in the symptoms of atopic dermatitis was observed, which persisted for up to 12 weeks without significant site reactions. Papp et al. also reported similar findings in another study [171]. Ruxolitinib cream was later approved by the FDA for the treatment of atopic dermatitis [172].

Furthermore, ruxolitinib has also displayed therapeutic benefits in the treatment of acute or chronic graft-versus-host disease (a/cGVHD) [173,174].

##### Metabolism

Shilling et al. [175] investigated the metabolic profile of ruxolitinib in healthy human subjects using [^14^C]-labeled ruxolitinib. The results revealed that the parent drug constitutes the major circulating component in plasma (58–74%). Ruxolitinib underwent extensive metabolism, mainly through oxidative pathways that occurred preferentially at the 2- or 3-position of the cyclopentyl ring and resulted in a series of hydroxy/oxo-metabolites, Figure 57. In addition, *O*-glucuronide conjugates (M28 and M51) were also identified. Among these metabolites, the 2-hydroxycyclopentyl derivative of ruxolitinib (M18) was the major one.

Shi et al. [176] evaluated the impact of the CYP3A4 inhibitor and inducer on the pharmacokinetics of ruxolitinib. The results of this study revealed an increase of the plasma concentration of ruxolitinib by the CYP3A4 inhibitors, ketoconazole, and erythromycin. On the other hand, a decrease in total ruxolitinib was observed on co-administration with the CYP3A4 inducer rifampin.

#### 1.7.10. Tofacitinib

##### Approval History

Tofacitinib (Figure 58) is a JAK inhibitor that was approved by the FDA for rheumatoid arthritis in 2012 [64,177,178]. It was also approved for the treatment of psoriatic arthritis and ulcerative colitis in 2017 and 2018, respectively [115]. In addition, tofacitinib has received FDA approval for the treatment of juvenile idiopathic arthritis in 2020 [179]. In December 2021, tofacitinib was approved for the treatment of active ankylosing spondylitis [180].

##### Synthesis

Tofacitinib was obtained from the reaction of compound **12** and (3*R*,4*R*)-1-benzyl-*N*,4-dimethylpiperidin-3-amine **112** [115,181]. Preparation of the substituted piperidine **112** could be prepared from different starting materials using diverse reaction conditions [64,182]. Among these materials, the 4-methylpyridin-3-amine **108** was reacted with dimethyl carbonate to give **109**, Figure 14. Rhodium-catalyzed hydrogenation of **109** gave **110**, which underwent reductive amination to give **111** as a racemic mixture. The (3*R*,4*R*)-1-benzyl-*N*,4-dimethylpiperidin-3-amine enantiomer of **111** was resolved using L-di-*p*-toluoyl-tartaric acid (L-DTTA) to give **112**.

The reaction of compound **12** with the piperidine derivative **112** afforded compound **113**, Figure 14. Removal of the benzyl group in **113** was achieved using Pd/C-catalyzed hydrogenation, which gave **114**. Tofacitinib was then obtained from the reaction of **114** with cyanoacetic acid 2,5-dioxopyrrolidin-1-yl ester **115**, while treatment of tofacitinib with citric acid afforded the citrate salt.

##### Target Kinases

The kinase inhibitory activity of tofacitinib was evaluated in several studies [83,96]. The results (Figure 59) revealed the inhibitory activity of tofacitinib against the four JAKs at IC_50_ values in the range of 1.1–42 nM, where the highest inhibitory activity was against JAK3 [96]. In addition, tofacitinib inhibited JAK1 and JAK2 at IC_50_ values of 2.9 nM and 1.2 nM, respectively.

On the other hand, the inhibitory activity of tofacitinib against JAKs was also evaluated by Clark et al. [83]. The results of that study revealed its inhibitory activity against JAK3 at IC_50_ of 55 nM. In addition, tofacitinib inhibited the enzymatic activity of JAK1 and JAK2 at IC_50_ values of 15.1 nM and 77.4 nM, respectively.

##### Crystal Structures

Tofacitinib exists as a co-crystallized ligand in five crystal structures. These crystals include tofacitinib in complex with JAK1 (pdb: 3EYG) [183], JAK2 (pdb: 3FUP) [183], JAK3 (pdb: 3LXK) [184], TYK2 (pdb: 3LXN) [184], and PRK1 (pdb: 4OTI) [185]. The binding interactions of tofacitinib into JAK1 (pdb: 3EYG) are depicted in Figure 60. Tofacitinib shows two conventional hydrogen bonds with Gly884 and Leu959.

The binding mode and interactions of tofacitinib with JAK2 (pdb: 3FUP) are depicted in Figure 61. Tofacitinib shows two conventional hydrogen bonds with Gly858 and Leu932. In addition, tofacitinib forms three carbon hydrogen bonds with Leu855, Arg980, and Asn981.

In addition, the binding mode, and interactions of tofacitinib with JAK3 (pdb: 3LXK) are visualized in Figure 62. Tofacitinib shows one conventional hydrogen bond with Leu905. In addition, tofacitinib displays four carbon hydrogen bonds with Leu828, Lys855, Arg953, and Asn954 amino acids in JAK3.

On the other hand, the orientation of tofacitinib into TYK2 (pdb: 3LXN) is visualized in Figure 63. Tofacitinib shows two conventional hydrogen bonds with Gly906 and Val981.

In addition, the binding mode/interactions of tofacitinib into PRK1 (pdb: 4OTI) are visualized in Figure 64. Tofacitinib shows one conventional hydrogen bond with Ser704 and one carbon hydrogen bond with Leu627.

##### Pharmacological Activities and Uses

Several clinical trials were performed to evaluate the efficacy of tofacitinib in the treatment of rheumatoid arthritis. In 2008, the results of a clinical trial (NCT00814307) of tofacitinib revealed improvement in the signs and symptoms of rheumatoid arthritis [186]. 

In another clinical trial (NCT00853385), tofacitinib showed similar efficacy to adalimumab in patients with rheumatoid arthritis [187]. In patients receiving methotrexate, tofacitinib also stopped the progression of structural damage [188]. By the end of 2012, tofacitinib received the first approval for the treatment of rheumatoid arthritis.

Since the first approval of tofacitinib in 2012 for rheumatoid arthritis, several studies have been performed to evaluate its efficacy in several types of inflammatory and immune diseases. In a clinical trial (NCT01882439), tofacitinib reduced active psoriatic arthritis in patients who had an inadequate response to TNF inhibitors [189]. Mease et al. also evaluated the efficacy of tofacitinib in patients with psoriatic arthritis who had an inadequate response to DMARDs [190].

In another clinical trial (NCT00787202), tofacitinib was investigated for the treatment of patients with severely active ulcerative colitis [191]. The results showed that clinical response and remission were more expected in the treated patients than in those receiving placebo. In addition, Huang et al. also reported an improvement in arthritis in a 13-year-old girl, with complete remission within three months [192]. Tofacitinib was approved by the FDA for the treatment of active psoriatic arthritis, ulcerative colitis, and juvenile idiopathic arthritis [177,178].

On the other hand, tofacitinib displayed inhibitory activity against LCK, which could also contribute to its pharmacological activities [64].

##### Metabolism

The pharmacokinetics parameters of tofacitinib were evaluated by Dowty et al. [193] using [^14^C]-labeled tofacitinib in healthy males. The results showed rapid absorption with the parent drug forming ~70% of the circulating activity in plasma.

Guo et al. investigated the metabolism of tofacitinib in vitro using a recombinant CYP3A4 enzyme [194]. Tofacitinib was incubated with mixed male human liver microsomes (HLMs) or individual human recombinant P450 enzymes. The results revealed the formation of a tofacitinib epoxide metabolite which was trapped by *N*-acetyl-L-cysteine (NAC), Figure 65. On the other hand, the metabolism of tofacitinib afforded also an *α*-keto-aldehyde metabolite, which was catalyzed by CYP3A4. This metabolite was trapped with *Nα*-acetyl-L-lysine.

#### 1.7.11. Upadacitinib

##### Approval History

Upadacitinib (Figure 66) is a JAK1 inhibitor that was approved to treat rheumatoid arthritis by the FDA in August 2019 [67]. It was also approved for the treatment of patients with psoriatic arthritis [195]. In 2022, upadacitinib was approved for treatment of atopic dermatitis [196]. In addition, it was approved in March 2022 to treat patients with moderate-to-severe active ulcerative colitis [197].

##### Synthesis

Upadacitinib was synthesized from the coupling of compounds **121** and **128** [104,198]. Compound **121** was obtained **116** in a three steps synthesis. In the first step, 3,5-dibromopyrazin-2-amine **116** was reacted with ethynyltrimethylsilane **117** to afford **118**, which was reacted with *p*-toluenesulfonyl chloride to give **119**. Palladium-catalyzed amination of **119** with ethyl carbamate **120** afforded **121**. Figure 15.

To prepare compound **128**, the ethyl acrylate **123** was first reacted with compound **122** to give **124**, which was then reacted with triflic anhydride to give **125**, Figure 15. The reaction of **125** with ethyl boronic acid gave **126**, which underwent alkaline hydrolysis to give **127**. Compound **128** was obtained from **127** in a three-step synthesis, which included ruthenium-catalyzed hydrogenation, reaction with carbonyldiimidazole (CDI), and reaction with trimethyl sulfoxonium chloride.

The coupling of **121** and **128** afforded **129**, which underwent intramolecular cyclization to give the tricyclic imidazo[1,2-*a*]pyrrolo[2,3-*e*]pyrazine **130**, Figure 15. Palladium-catalyzed hydrogenation of **130** was performed to remove the protecting group, followed by treatment with hydrochloric acid to give the salt **131**. The reaction of **131** with CDI and trifluoroethylamine afforded upadacitinib.

##### Target Kinases

The JAKs inhibitory activity of upadacitinib was evaluated by Parmentier et al. [199]. The results revealed the highest inhibitory activity against JAK1 (IC_50_ = 47 nM), Figure 67. The study also revealed inhibitory activity for upadacitinib against JAK2 at IC_50_ 120 nM, which indicates 2.5-fold lower inhibitory activity compared to JAK1.

In addition, the results of the kinase inhibitory assay of upadacitinib against 70 kinases revealed weak inhibitory activity against kinases other than JAKs [199].

##### Crystal Structures

Upadacitinib has not yet been reported in a crystal structure with any of its target kinases.

##### Pharmacological Activities and Uses

Upadacitinib is a second-generation selective JAK inhibitor that was evaluated for the treatment of different types of inflammatory and immune diseases [200,201,202,203]. Evaluation of the efficacy of upadacitinib in the treatment of rheumatoid arthritis revealed a fast and favorable efficacy profile [202]. In addition, Smolen et al. [204] evaluated the efficacy of upadacitinib monotherapy in rheumatoid arthritis. The results also revealed significant therapeutic outcomes compared to methotrexate. The medical use of upadacitinib, either alone or in combination therapy for the treatment of rheumatoid arthritis, was associated with lower direct medical costs [203]. In August 2019, upadacitinib was approved for the treatment of moderate-to-severe rheumatoid arthritis.

Moreover, several clinical trials were also performed to evaluate the efficacy of upadacitinib in the treatment of psoriatic arthritis. In a clinical trial (NCT03104400.) for treatment of psoriatic arthritis, upadacitinib produced a rapid and sustained improvement in patient outcomes [201]. Upadacitinib was also evaluated in a 24-week, phase 3 trial to treat psoriatic arthritis [205]. The results revealed a significantly higher number of patients with ACR20 compared to the placebo. In addition, upadacitinib at a daily dose of 30 mg showed superior results to those of adalimumab. Furthermore, no significant safety signals were observed when upadacitinib was evaluated in patients with psoriatic arthritis. Burmester et al. [206] also evaluated the safety of upadacitinib in patients with psoriatic arthritis for up to 3 years, where the results revealed a safety profile similar to that observed in rheumatoid arthritis. Currently, upadacitinib has been approved by the FDA and EMA for the treatment of patients with active psoriatic arthritis [195,207].

Furthermore, upadacitinib was also studied in adult patients with atopic dermatitis, where the results showed superior efficacy compared to the human monoclonal antibody dupilumab [208]. In January 2022, upadacitinib was also approved by the FDA to treat refractory, moderate to severe atopic dermatitis in children aged ≥ 12 years.

##### Metabolism

The results of the in vitro metabolic study of upadacitinib suggested that it undergoes metabolism by cytochrome P450 [209]. On the other hand, the contribution of CYP2D6 in the metabolism of upadacitinib was very minor [209]. Coadministration of upadacitinib with the CYP3A4 inhibitor, ketoconazole, resulted in a weak effect on its concentration [210]. However, coadministration of upadacitinib with rifampin, a broad CYP inducer, resulted in a decrease in upadacitinib concentration by ~50% [211].

In conclusion, the target kinases of the eleven JAK inhibitors and the approval data including the approval date and the disease for which these inhibitors were approved are presented in Table 1.

## 2. Conclusions

In the current review, eleven of the JAK inhibitors that received approval for the treatment of inflammatory, autoimmune, and myeloproliferative neoplasms were discussed. These drugs are abrocitinib, baricitinib, delgocitinib, fedratinib, filgotinib, oclacitinib, pacritinib, peficitinib, ruxolitinib, tofacitinib, and upadacitinib. The synthetic routes of these drugs, including the original and/or alternative pathways, were described. The crystal structures of these drugs in various kinases were also listed. Their binding modes and interactions were visualized, where two key hydrogen-bonding interactions were observed with Leu959 and Leu932 in JAK1 and JAK2, respectively. Furthermore, the kinase inhibitory activities and pharmacological uses of the eleven drugs were also summarized. Based on their inhibitory activity against the target kinases, these drugs could be classified as either selective or nonselective JAK inhibitors. Among these drugs, several JAK1 inhibitors have been approved for the treatment of inflammatory and autoimmune conditions. On the other hand, the drugs approved for treatment of myeloproliferative neoplasms target JAK2 and its mutant form (JAK2V617F). In addition, the metabolic studies of the eleven drugs revealed the formation of several oxidation metabolites, which were mediated by CYP450 enzymes. On the other hand, few conjugation metabolites such as glucuronide acid and sulfate conjugates were detected among the metabolites of filgotinib and peficitinib, respectively. To sum up, the data in this review may assist in the design of new JAK inhibitors with potential therapeutic benefits.

## 3. Perspective

The high efficacy of the clinically approved JAK inhibitors in the treatment of inflammatory and autoimmune diseases has attracted much attention. However, most of them are non-selective inhibitors, which may account for some of their adverse effects, such as anemia, thrombocytopenia, upper respiratory tract infection, and herpes infection [212,213,214,215,216]. Accordingly, design and development of new potent, selective, and more safe JAK inhibitors could provide a solution to these adverse effects [217].

Currently, several nonselective JAK inhibitors, including brepocitinib, cerdulatinib, gusacitinib, and momelotinib (Figure 68), are being investigated for their efficacy in inflammatory and cancer diseases [218,219,220,221]. In addition, the efficacy and safety of several selective JAK inhibitors are being investigated in clinical trials. Among these inhibitors, the selective JAK1 inhibitor, itacitinib showed promising efficacy when evaluated in patients with aGVHD [222].

The JAK2 inhibitor, gandotinib (Figure 68), showed high potency toward the JAK2V617F mutation and showed promising potential in the treatment of myeloproliferative disorders [223]. In addition, JAK2/FLT3 dual inhibitors could provide better therapeutic option for acute myeloid leukemia [30,224]. Currently, it is seeking FDA approval for the treatment of myelofibrosis. On the other hand, the JAK3 inhibitor, decernotinib also showed a high potential activity in the treatment of rheumatoid arthritis [225].

Several JAK inhibitors have been reported to bind reversibly to allosteric sites in JAKs [34,42]. Among these inhibitors, deucravacitinib and LS104 are being investigated in clinical trials. This type of JAK inhibitor could provide advantages over the currently used ATP-competitive inhibitors [27]. However, none of these allosteric inhibitors has been approved for clinical use. Future research in this area could lead to approval of the allosteric inhibitors for clinical use.

Development of irreversible JAK3 inhibitors that can bind covalently with the unique Cys909 reside has attracted a great attention in the last few years [43]. Several small molecules have displayed potent and selective inhibition of JAK activity [43,44,226]. The design of new JAK3 covalent inhibitors could be supported by the success of ritlecitinib in reaching the clinical trial (NCT04517864). In addition, advances in X-ray and covalent docking may also play a crucial role in the design of this type of JAK inhibitors.

Recently, JAK inhibitors showed promising potential in the treatment of COVID-19-related cytokine storm [227]. However, among several drug combinations evaluated in the treatment of COVID-19 [228,229], the combination of baricitinib plus remdesivir has received EUA by the FDA [92,230,231]. Moreover, the clinical trials of tofacitinib (NCT04469114) and nezulcitinib (NCT04402866) are being performed in patients with COVID-19 related lung problems [36,232]. Although these results also support the investigation of other JAK inhibitors in the treatment of COVID-19 related problems, the adverse effects of these inhibitors on the immune responses must be evaluated [233].

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
