# Peer review of "A Comprehensive Overview of Globally Approved JAK Inhibitors"

_pharmaceutics, 2022, doi:10.3390/pharmaceutics14051001_

Round 1

Reviewer 1 Report

In the current review, ten of the JAK inhibitors, that received approval for clinical use have been discussed. These drugs are abrocitinib, baricitinib, delgocitinib, fedratinib, filgotinib, oclacitinib, peficitinib, ruxolitinib, tofacitinib, and upadacitinib. The synthetic routes of the ten drugs were described. In addition, their inhibitory activities against different kinases and their pharmacological uses have also been explained. Moreover, their crystal structures with different kinases were summarized with a primary focus on their binding modes and interactions. The proposed metabolism of the ten drugs were also illustrated. To sum up, the data in the current review could help in the design of new JAK inhibitors with potential therapeutic benefits in inflammatory and autoimmune diseases. This paper well written and in my opinion paper should be accepted after minor revision.

Comments to the Author:

  1. The introduction should detail the aim and goals of the study by referring to the available literature studies and explaining how the undertaken studies would make these goals possible. It is not clearly expressed how the database is expected to show inflammatory and autoimmune diseases activity.
  2. Author should also explore which JAK is a family of cytoplasmic non-receptor tyrosine kinases is most impotent for activity.
  3. Figure 1 and 2 is not clear, it should be revise in high resolution
  4. It is noted that the manuscript needs careful editing in English spelling and grammar.
  5. Conclusion: This part is also written poorly. The conclusion should include the most important findings and the outcomes thereof.

Author Response

Response to reviewers’ comments

Reviewer 1

Comments and Suggestions for Authors

In the current review, ten of the JAK inhibitors, that received approval for clinical use have been discussed. These drugs are abrocitinib, baricitinib, delgocitinib, fedratinib, filgotinib, oclacitinib, peficitinib, ruxolitinib, tofacitinib, and upadacitinib. The synthetic routes of the ten drugs were described. In addition, their inhibitory activities against different kinases and their pharmacological uses have also been explained. Moreover, their crystal structures with different kinases were summarized with a primary focus on their binding modes and interactions. The proposed metabolism of the ten drugs were also illustrated. To sum up, the data in the current review could help in the design of new JAK inhibitors with potential therapeutic benefits in inflammatory and autoimmune diseases. This paper well written and in my opinion, paper should be accepted after minor revision.

We highly appreciate the valuable comments of reviewer 1 and the valuable suggestions and corrections that have emerged after his careful and precise revision, which would help in improving the quality of the manuscript. In addition, we indicated the revisions/corrections with a yellow highlighter in the revised manuscript. Below, are our responses to the comments, point-by-point. We will therefore reply to all points one by one as follows

Comments to the Author:

Comment 1:

The introduction should detail the aim and goals of the study by referring to the available literature studies and explaining how the undertaken studies would make these goals possible. It is not clearly expressed how the database is expected to show inflammatory and autoimmune diseases activity.

Response:

We would like to thank reviewer 1 for this valuable comment.

  • The aim of the current review was added to the introduction part
  • In addition, the sub-section “3. (Therapeutic potential of JAKs“ in the introduction was revised and corrected. In this section, the types of JAKs involved in the treatment of inflammatory and myelofibrosis neoplasms are indicated.

Comment 2: Author should also explore which JAK is a family of cytoplasmic non-receptor tyrosine kinases is most impotent for activity.

Response:

  • The importance of targeting JAK1 and JAK2 with small-molecular inhibitors in the treatment of inflammatory, autoimmune, and myelofibrosis  neoplasms are indicated in the revised “3. Therapeutic potential of JAKs”.

Comment 3: Figure 1 and 2 is not clear, it should be revise in high resolution

Response: Figures 1 and 2 were revised and updated in high resolution in the revised manuscript.

Comment 4: It is noted that the manuscript needs careful editing in English spelling and grammar.

Response: We would like to thank review #2 for this comment. The manuscript was revised for any typo and grammatical mistakes.

Comment 5: Conclusion: This part is also written poorly. The conclusion should include the most important findings and the outcomes thereof.

Response: The conclusion was revised and corrected in the revised manuscript. It was updated with short and concise statements describing the conclusions of this study.

Reviewer 2 Report

Shawky et al report on a review of globally approved JAK inhibitors. These drugs are used in the therapy of some autoimmune diseases as well as in haematological malignancies. The paper is well written and organised, containing useful information. However several points could be addressed:

Major points:

  1. The authors are focused on the drugs approved for autoimmune diseases. In fact, in the Figure 3 no reference to ruxolitinib or fedratinb is included. This must be modified
  2. A number of comparative tables including all the JAK inhibitors already approved will facilitate the lecture of the paper. These tables can include: indications, previous clinical trials, secondary effects. The paper is focused in chemist while clinical information is missing.
  3. Lack of information regarding new JAK inhibitors, mainly pacritinb.
  4. Some repetitions should be avoided (as an example: pag 37 and 42 regarding ruxolitinib indications)

Minor points

In order to shorten the current pages number (54 excluding references) some figures can be included as Sup Material.

Author Response

Reviewer 2

Comments and Suggestions for Authors

Shawky et al report on a review of globally approved JAK inhibitors. These drugs are used in the therapy of some autoimmune diseases as well as in haematological malignancies. The paper is well written and organised, containing useful information. However several points could be addressed:

We highly appreciate the reviewer’s comments on this work. We also appreciate his valuable corrections that have emerged after careful and precise revision which would help in improving the quality of the manuscript. The corrections made in the final manuscript were indicated by a yellow highlighter. Below, we will reply all points one by one as follows:

Major points:

Comment 1: The authors are focused on the drugs approved for autoimmune diseases. In fact, in the Figure 3 no reference to ruxolitinib or fedratinb is included. This must be modified

Response: We would like to thank the reviewer for this correction. This part was revised and corrected. The missing references were added to the revised manuscript.

Comment 2: A number of comparative tables including all the JAK inhibitors already approved will facilitate the lecture of the paper. These tables can include: indications, previous clinical trials, secondary effects. The paper is focused on chemist while clinical information is missing.

Response: We thank the reviewer for this valuable suggestion. The primary aim of this review was to provide the readers with an integrated overview of the chemical and pharmacological data of the globally approved JAK inhibitors. However, we have added Table 1 in the revised manuscript which includes the eleven drugs with their primary target JAKs, the disease for which these inhibitors were approved, and the approval data including the date and the necessary references. In addition, some of the clinical trials of these drugs were also added.   

Comment 3: Lack of information regarding new JAK inhibitors, mainly pacritinib.

Response: The data of the recently approved JAK inhibitor, pacritinib, including the approval history, synthesis, target kinases, crystal structure, pharmacological uses, and metabolism, were included in the revised manuscript. Accordingly, 11 (eleven) drugs were discussed in the updated manuscript. The number of figures, schemes, and references were modified following this update.

Comment 4: Some repetitions should be avoided (as an example: pag 37 and 42 regarding ruxolitinib indications)

Response: The repeated indications of ruxolitinib were revised and corrected in the revised manuscript.

Comment 5: Minor points. In order to shorten the current pages number (54 excluding references) some figures can be included as Sup Material.

Response: We thank the reviewer for this valuable suggestion. However, we think that it would be easier for the readers if all the figures were included in the manuscript.

Round 2

Reviewer 2 Report

No further comments